# Characterization of Volatile Compounds of *Rosa roxburghii* Tratt by Gas Chromatography-Olfactometry, Quantitative Measurements, Odor Activity Value, and Aroma Intensity

**DOI:** 10.3390/molecules26206202

**Published:** 2021-10-14

**Authors:** Yunwei Niu, Ruolin Wang, Zuobing Xiao, Xiaoxin Sun, Pinpin Wang, Jiancai Zhu, Xueying Cao

**Affiliations:** School of Perfume and Aroma Technology, Shanghai Institute of Technology, Shanghai 201418, China; yunweiniu@163.com (Y.N.); wangruolin82@163.com (R.W.); Phoebe1397@163.com (X.S.); wang_pinpin@163.com (P.W.); zjc01@163.com (J.Z.); cxy13023330303@163.com (X.C.)

**Keywords:** *Rosa roxburghii* tratt, characteristic compounds, GC–MS–O, odor threshold, key note, omission tests

## Abstract

*Rosa roxburghii* tratt (RRT), widely distributed in the southwest of China, is favored by consumers for its good taste and healthy functions. In this study, thirty-seven compounds of *Rosa roxburghii* Tratt (RRT) were identified and quantified by gas chromatography–olfactometry (G–O) and gas chromatography–mass spectrometry (GC–MS) analysis. Furthermore, ethyl 2-methylpropanoate, ethyl butanoate, ethyl 2-methylbutyrate, and ethyl hexanoate were present with much higher odor activity values (OAVs) than other compounds. The key notes were confirmed by omission tests. Possible interaction among key notes was investigated through odor intensity determination and sensory analysis. It showed fruity and woody notes had synergistic effects. Full factorial design was used to evaluate the notes contribution to the whole odor. One important finding is the major effect of order interactions, fruity note (X1) and woody note (X4) especially, emphasizing the existence of complex interactions occurring between odor notes. The interaction X1X4 was further investigated. The woody note has a positive effect when the fruity note is also in the mixture but tends to show a negative effect otherwise.

## 1. Introduction

Aroma is an important characteristic of food and plays an important role when assessing food quality. Tens of thousands of natural aroma volatiles exist and contribute to different food aromas; therefore, it was considered that food aroma was the combination of different aroma volatiles and relative amounts.

*Rosa roxburghii* tratt (RRT), which belongs to the rosaceae family, is widespread in the southwest of China. Its juice has been made as beverage and herbal tea in folk for the functions of tonifying spleen, cuing diarrhea, and their good tastes [1]. *Rosa roxburghii* tratt is an important commercial horticultural crop in China that is recognized for its nutritional and medicinal values. In particular, Guizhou, Sichuan, Yunnan, southern Shaanxi, Hubei, and Hunan have a large area and a large output. The actual output is about 1200 tons. Recently, studies on *Rosa roxburghii* tratt have focused on the antioxidant [2,3], chemical [4], biological properties [5], and pharmacological properties [6] of the fruit. There are no reports aiming to investigate the aroma compounds of RRT fruits.

RRT odors are composed of a large number of volatile compounds, and only a small fraction contributes to their global odor. The gas chromatography–olfactometry (GC–O) method, proposed by Fuller et al. as early as 1964 [7], couples traditional gas chromatographic analysis with sensory detection in order to study complex mixtures of odorous substances and to identify odor active compounds. The GC–O technique is already widely used for the evaluation of food aromas, such as banana, pineapple, mango, etc. [8]. In mixtures, the diversity of sensory perceptions reported result from qualitative (odor quality) and quantitative (odor intensity) perceptual interactions between odorants, defined in various ways by different authors [9]. For instance, the technique was conducted through comparing the overall perceived intensity of a mixture to the intensities of the components smelled alone [10]. Five outcomes can be found in the mixture, such as complete addition, hyper-addition, partial addition, compromise, and compensation. Niu et al. investigated sensory interaction between esters made the odor quality of light aroma-type liquor outstanding by the aroma quality and aroma intensity in binary ester mixtures [11]. From the σ/τ plot, hypo-addition action was frequent in binary mixtures, and hyper-addition action occurred at low level intensity (generally τ < 0.5). Level independence was not observed in studied five binary mixtures. Xiao et al. focused on the impact of esters on the perception of floral aroma in rose essential oil [12]. The floral reconstitution in alkanes solution was supplemented with the five esters at high, medium, and low concentration and then analyzed by quantitative descriptive analysis. It was revealed that ethyl octanoate, ethyl tetradecanoate, and citronellyl acetate add overall aroma, and geranyl acetate masks overall aroma perception in a model floral mixture. Sensory profiles highlighted changes in the perception of aroma nuances in the presence of the five esters, with specific perceptive interactions, and reported on the graph based on two parameters (σ = f(τ)).

Taking into account interactions between odorants in mixture requires the use of a specific experimental design. The most suitable methodologies involve the use of factorial designs. Hallier et al. proposed a fractionated factorial design to evaluate the impact of five odor families by omission tests, allowing the estimation of main effects and first order interactions [13]. Paravisini et al. hypothesized that caramel odor was the result of complex interactions between odor notes; it also implied the use of a full factorial design to evaluate high-order interactions [14]. This method can verify the interaction between several aroma substances more scientifically and effectively. Thus, a 24 factorial design was built to study the interactions among the four more relevant odor notes in mixtures. Thirty commercial orange juice samples were evaluated by descriptive sensory analysis using a 15 point scale and GC–O [15].

The aims of this study were to (a) identify the key odor-active compounds in RRT samples by GC–O, GC–MS, and calculation of OAVs of volatile compounds; (b) confirm the key notes; (c) use full factorial design to evaluate the notes’ contribution to the whole odor; (d) investigate the possible interactions among key notes through odor intensity determination and sensory analysis.

## 2. Results and Discussion

### 2.1. Identification and Quantitation of Compounds and OAV Analysis

The compounds were identified by comparison of retention index (RI), odor descriptors with authentic standards, as shown in Table 1(a). In this study, 37 odorant compounds were detected after the GC–O analysis of RRT samples. The different AIs of the volatile compounds in each of the samples were mainly induced by concentration differences of these compounds. The AIs of the compounds ranged from 0.2 to 5. Ethyl butanoate exhibited the highest AI in RRT samples.

Esters were shown to be the largest class of aroma compounds in the RRT fruit samples. Thirteen esters were identified, which are summarized in Table 1(a). These included ethyl acetate, ethyl 2-methylpropanoate, ethyl butanoate, ethyl 2-methylbutanoate, 3-methyl butylacetate, ethyl hexanoate, ethyl 2-methylcrotonat, ethyl 3-hexenoate, ethyl heptanoate, ethyl E-2-hexenoate, etheyl octanoat, ethyl benzoate, and ethyl cinnamate. Esters were generally associated with fruity notes in the sensory descriptions from panelists. These esters were widespread in many fruits [16,17,18]. Higher AI values, presented in four compounds, ethyl 2-methylpropanoate, ethyl butanoate, ethyl 2-methylbutanoate, and ethyl hexanoate, which were the most powerful odor-active compounds contributing to the aroma profile of RRT fruit, were identified to be primarily responsible for aroma in several cultivators.

In addition to ester compounds, acids were another important class of aroma active compounds found in the RRT samples. Seven acids were identified, which are summarized in Table 1(a). These included acetic acid, 3-methylbutanoic acid, hexanoic, E-3-hexenoic acid, heptanoic acid, and octanoic acid. According to the AI size of the aroma substance, acetic acid, 3-methylbutanoic acid, and hexanoic acid were the most powerful odor-active compounds contributing to the aroma profile of RRT fruits.

The concentrations and odor activity values (OAVs) of the ester compounds obtained by SPME–GC–MS are presented in Table 1(b).

A total of 37 volatile compounds in RRT juice samples were quantitated (Table 1(b)). The major volatile compounds of samples were ethyl acetate (45.24 mg/kg), acetic acid (88.72 mg/kg), heptanoic acid (97.12 mg/kg), and 3-methylbutanoic acid (8.69 mg/kg). The contributions of volatile compounds in the samples not only depend upon the amounts of each compound but also their odor threshold value. According to results obtained by Guth, those with OAVs greater than 1 were considered to contribute to the aroma of samples [19]. In addition to ethyl acetate, the OAVs of the other compounds were greater than 10 in the samples and are therefore considered as aroma-active compounds in RRT fruits. Among these compounds, ethyl 2-methylpropanoate (0.12 mg/kg), ethyl butanoate (0.59 mg/kg), ethyl 2-methylbutyrate (0.24 mg/kg), 3-methylbutyl acetate (0.19 mg/kg), E-2-hexenal (1.78 mg/kg), ethyl hexanoate (2.21 mg/kg), 3,7-dimethyl-1,3,6-Octatriene (0.49 mg/kg), 2-Heptanol (1.06 mg/kg), ethyl heptanoate (0.02 mg/kg), ethyl E-2-hexenoate (0.04 mg/kg), and etheyl octanoat (0.54 mg/kg) were present at relatively low concentrations (<1 mg/kg). Despite their low levels, the OAVs of these compounds were above 10, so these compounds might significantly contribute to the aroma of RRT fruits [20]. Ethyl 2-methylpropanoate (OAV of 1167), ethyl butanoate (OAV of 3279), ethyl 2-methylbutyrate (OAV of 811), and ethyl hexanoate (OAV of 2205) were present with much higher OAVs than other compounds.

However, volatile compounds such as 2-heptanone, ethyl tiglate, ethyl 3-hexenoate, 2-nonanone, nonanal, benzaldehyde, 2-undecanone, caryophyllene, butyric acid, heptanoic acid, octanoic acid, and eugenol had high AIs, but they do have a lower OAV, which could be related to the difference of thresholds in air and water. According to previous studies, OAV might be a more effective method for the verification of the aroma-active compounds [21]. The GC–O method simply verifies the contribution of each compound to the overall aroma, but the OAV method considers the interaction for the aromatic compound and the food matrix. Thus, the use of OAVs may provide a better assessment of the OAVs of compounds as long as the quantitative data and the odor thresholds detected are accurate.

### 2.2. Omission Tests

The omission tests were used to deeply investigate the aroma contribution of the odor notes. This strategy was used to determine whether combinations of several volatile compounds could have an effect on the global odor extract characteristics. Multiple omissions were probably the best method to obtain meaningful results with a minimum of repetitive testing [22]. The model RRT juice was used to complete the omission tests to confirm the key compounds [23] that had a strong influence on the perception of aroma.

According to Table 2, a total of six omission models, in which a single note was omitted, were prepared to complete the omission experiments. The omission models were compared with TAR by a triangle test according to previous study [24].

The model without fruity or sour notes showed a high significant difference (α ≤ 0.01) compared to the aroma of TAR. Furthermore, panel members could also detect a significant difference (α ≤ 0.05) between green and woody notes. These data indicated that fruity, sour, green, and woody notes could be responsible for the typical aroma of RRT samples. Hallier et al. (2004) evaluate the impact of five odor families of Silurus glanis by omission tests [13]. This fact is not totally surprising, as these compounds were present in the mixture at high concentration, well above its olfactory threshold. In some studies, similar results were observed using mixtures involving pyridine and linalool, linalyl acetate, or lavender essential oil, where the smell of the compound with the highest intensity predominated in the mixture, completely masking the smell of the less intense compound in some cases [10].

### 2.3. Odor Intensity of Binary Mixtures

To analyze the quantitative olfactory interactions in four odor notes further, in this research, synthetic representation σ = f (τ) was used. Data concerning these different pairs were shown in Figure 1. It can be observed that mixture “3” (fruity and woody) lies in the hyper-addition area. It showed fruity and woody had a synergistic effect. This is a unique pair perfect addition, and synergy was restricted to some very exceptional cases. For instance, cases of hyper-addition were noticed previously by Laing [25]. Hyper-addition could occur when mixing low iso-intense fruity and woody odors [26]. For the six binary mixtures tested in this study, five pairs (1. fruity and sour, 2. fruity and green, 3. fruity and woody, 4. sour and green, 5. sour and woody, 6. woody and green) were in the partial addition area and more than half (83%) showed partial addition level. These results were agreement with previous studies which showed that partial addition was the most likely outcome, and the intensity is never less than the intensity of the weaker compounds [27].

### 2.4. Factorial Design

Assessing the typicality of the odor is a direct and efficient way to study the impact of odor categories. It is a straightforward evaluation of the positive or negative impact of the category. Thus, this experimental approach seems promising to understand more deeply the contribution of the volatile fraction to the odor of complex products [14]. Full factorial design was used to evaluate the notes contribution to the whole odor were shown in Figure 2. Among the six odor notes, only four were selected to keep the number of mixture in a manageable range for sensory evaluations (X1 = fruity; X2 = sour; X3 = green; X4 = woody). Fatty and floral notes were dropped due to both no significant impact of the blend. Relative contributions of main effects and interactions are shown on the Pareto charts. One important finding is the major effect of order interactions (X1X4 especially), emphasizing the existence of complex interactions occurring between odor notes.

As shown in Figure 3, The interaction X1X4 was further investigated. The woody note (X4) has a positive effect when the fruity note (X1) is also in the mixture but tends to show a negative effect otherwise.

## 3. Materials and Methods

### 3.1. Chemicals

Authentic standards were obtained from the following sources. Ethyl acetate (≥97%), ethyl 2-methylpropanoate (≥97%), ethyl butanoate (≥97%), ethyl 2-methylbutanoate (≥97%), 3-methylbutyl acetate (≥97%), 2-heptanone (≥97%), 3-methyl-1-butanol (≥97%), E-2-hexenal (≥97%), ethyl hexanoate (≥97%), ethyl 2-methylcrotonat (≥97%), 3,7-dimethyl-1,3,6-Octatriene (≥97%), phenyl ethylene (≥97%), ethyl 3-hexenoate (≥97%), 2-heptanol (≥97%), ethyl heptanoate (≥97%), ethyl E-2-hexenoate (≥97%), hexanol (≥97%), 2-nonanone (≥97%), nonanal (≥97%), etheyl Octanoat (≥97%), acetic Acid (≥97%), 2-nonanol (≥97%), benzaldehyde (≥97%), 2,6,6,10-tetramethyl-1-oxaspiro[4.5]dec-9-ene (≥97%), 2-undecanone (≥97%), caryophyllene (≥97%), butyric acid (≥97%), 3-methylbutanoic acid (≥97%), ethyl benzoate (≥97%), hexanoic acid (≥97%), α-ionone (≥97%), α-iononol (≥97%), E-3-hexenoic acid (≥97%), heptanoic acid (≥97%), octanoic acid (≥97%), ethyl cinnamate (≥97%), eugenol (≥97%), sucrose (99%), fructose (99%), glucose (99%), malic acid (99%), sodium chloride (≥ 99.5%), and a homologous series of alkanes (C6–C30) were purchased from Sigma-Aldrich (St. Louis, MO, USA). All of them were analytical reagents. Distilled water was purchased from Shanghai Titan Technology Co., Ltd. (Shanghai, China).

### 3.2. Materials

The volatile compounds of RRT fruit were studied. The fruits were harvested from *Rosa roxburghii* tratt fruit field of Guizhou city on 10 September 2018, which is its ripening stage. According to color, firmness, aroma, and the judgment of local growers, ripe fruits with similar size and without visible external damage were selected for analysis.

RRT fruits were washed with distilled water and squeezed into juice by a kitchen blender. Then, the fruit juices were filtered through a four-layer cheese cloth. Then, the filtered juice was immediately employed in the next experiment.

### 3.3. Headspace-Solid-Phase Microextraction (HS-SPME) Absorption of Aroma Compounds

The manual SPME holder, together with 20 mL vials, Teflon covers, and one 50/30 µm divinybenzene/carboxen/polydimethylsiloxane (DVB/CAR/PDMS) fiber were purchased from Supelco, Inc. (Bellefonte, PA, USA). The fiber was preconditioned for 20 min at 250 °C to make sure no residue remained before chemical adsorption. The main parameters, such as fiber, extraction time, extraction temperature, sample volume, and stirring speed, were investigated. Optimized SPME experimental conditions were established, according to the results obtained, i.e., a sample of 8 g and stirring speed of 80 rmp, 45 min of extraction time at 50 °C. Therefore, 8 g of fresh RRT juice, 1.5 g of sodium chloride, and 15 µL 2-octanol (400 mg/L, internal standard) were immediately transferred to the vial. Then, the vial was put into a thermostatic water bath. The fiber was exposed to the headspace of the sample (about 1 cm above the liquid surface) for 45 min at 50 °C with stirring speed of 80 rmp and then introduced to the GC injector with 5 min for desorption and analysis. Each RRT juice sample underwent the same procedure which was described above.

### 3.4. Gas Chromatography-Olfactometry (GC–O)

GC–O was performed on an Agilent 7890 GC coupled to an olfactory detection port Gerstel ODP-2. GC effluent was split 1:1 between the flame ionization detector (FID) and sniffing port. Samples were separated on both a HP-Innowax analytical fused silica capillary column (60 m × 0.25 mm × 0.25 μm; Agilent, Santa Clara, CA, USA) and a DB-5 analytical fused silica capillary column (60 m × 0.25 mm × 0.25 μm, Agilent, Santa Clara, CA, USA). The flow rate of carrier gas (hydrogen) was 2 mL/min. The oven temperature was first increased from 40 °C (6 min), at 3 °C /min, to 100 °C and then ramped at 5 °C min^−1^ to 230 °C (20 min); the injector and FID temperatures were set at 250 °C and 280 °C, respectively [26]. The temperature of the sniffing port was set to 250 °C, and the length between the Y-splitter and the sniffing port was 107 cm. Moist air was pumped into the sniffing port at 50 mL min^−1^ to provide comfort to the panelists.

A panel of fifteen trained panelists was recruited to perform GC–O analysis. The panelists were trained for 3 months in GC–O, using at least 30 odor-active reference compounds in a concentration 10 times above their odor thresholds in air. During a GC run described above, a panelist placed his/her nose close to the sniffing port, responded to the aroma intensity of the stimulus, and recorded the aroma descriptor and intensity value as well as retention time. The sniffing time of each run was not more than 30 min. The first panelist sniffed for 30 min and then the next panelist sniffed for 30 min. In this way, olfactory fatigue was avoided, and no other odors were generated in the interim. The aroma descriptors were determined by an evaluation of the odor quality of reference odorants previously. A six-point scale ranging from 0 to 5 was used for aroma intensity (AI) judgment: 0 = none, 1 = very weak, 2 = weak, 3 = moderate, 4 = strong, and 5 = very strong. The AI was an average result of the fifteen panelists [28].

### 3.5. Gas Chromatography–Mass Spectrometry (GC–MS)

Compounds were analyzed by an Agilent 7890 gas chromatograph (GC) system coupled with a 5973C mass spectrometer (MS). DB-5 analytical fused silica capillary column (60 m × 0.25 mm × 0.25 μm, Agilent, Santa Clara, CA, USA) was used for chromatographic separations. The flow rate of the carrier gas helium was 1.0 mL/min. The MS parameters included electron impact ionization with electron energy of 70 Ev and mass range of *m*/*z* 30–450; initial oven temperature was 40 °C. After holding for 6 min, the oven temperature ramped to 100 °C at a rate of 3 °C /min and then increased to 230 °C at a rate of 5 °C/min with a 20 min hold. The volatile compounds were determined by authentic standards, retention indices (RIs) and Wiley7n.l Database (Hewlett-Packard, Palo Alto, CA, USA). For calculation of RI, a C6-C30 n-alkanes series (concentration of 1000 mg/L in n-hexane) from Sigma-Aldrich was used.

For the preparation of the *Rosa roxburghii* tratt juice model solution (RJMS), the composition was 0.6 g sucrose, 2.1 g fructose, 1.4 g glucose, and 1.3 g malic acid in 100 g of water. The quantification of aroma compounds was performed using standard curves according to the method of reference that was described previously [29]. The calibration curves are shown in Table 1(b), where y represented the peak area ratio (peak area of volatile standard/peak area of internal standard), and x represented the concentration ratio (concentration of volatile standard/concentration of internal standard). The extraction method of the standard volatiles for making the standard curve was as same as the sample extraction method. The experiments were performed in triplicate.

### 3.6. Odor Activity Values (OAV)

The odor activity values of compounds in RRT juice were measured as the ratio of the concentration of each compound to its detection threshold in water. Threshold values were taken from the literature [30,31].

### 3.7. Sensory Analyses

#### 3.7.1. General Conditions

Sensory analyses were performed as described by Martin and de Revel [32]. Samples were evaluated at controlled room temperature (20 °C) at least 12 h in individual booths, using covered, black ISO glasses, containing about 50 mL of liquid, coded with three-digit random numbers. Sessions lasted approximately 5 min.

#### 3.7.2. Sensory Panel

The panel consisted of 15 judges, 7 males and 8 females, aged 24–45. All panelists were research laboratory staff at School of Perfume and Aroma Technology, Shanghai Institute of Technology.

#### 3.7.3. Descriptive Sensory Analysis

The RRT juice was evaluated by Panel. First, 5 g of RRT juice was prepared in a 20 mL vial covered with Teflon and subjected to panelists without peculiar smell at 25 °C. Then, the panelists discussed aroma compositions of the RRT juice. Subsequently, the organoleptic characteristic descriptors were quantified using six sensory attributes (“Sour”, “Fruity”, “Green”, “Fatty”, “Woody”, and “Floral”). The score of each sample was presented on the basis of a 10 point scale (0, none; 5, moderate; and 10, very strong). The whole experiment was replicated in triplicate by each panelist.

#### 3.7.4. Omission Experiments

To obtain the key notes of RRT, six omission models were prepared. Based on the quantitative results, all the compounds were mixed and added into RJMS to prepare the total aromatic reconstitution (TAR). Each omission model was compared with TAR by triangular tests as described in previous study [33]. All the tested samples were determined by the panel and arranged in a random code (three repetitions). The panelists were required to sniff the samples and identify the different one.

#### 3.7.5. Determination of Aroma Intensity of Binary Mixtures of the Notes

Six groups of binary mixtures of the key notes (fruity, sour, green, and woody) were studied. They were mixed in the ratio of their quantitative concentrations of RRT (Table 1(b)). Prior to measurement, three stimuli were tested by the panel for each mixture. The panel was trained to familiarize the subjects and memorize the intensity references. Each stimulus was evaluated in three times using an 11 point interval scale (0 = none, 10 = extra strong). A mixture of all compounds of RRT with five concentration levels was selected as an intensity reference. The intensity was identified by panelists who were not told whether or not stimuli were mixtures.

Statistical data were analyzed using the Wilcoxon signed-rank statistical nonparametric test (XLSTAT software). All descriptors are mean-centered for each panelist and scaled to unit variance. The statistically significant level was 5% (*p* < 0.05). Experimental data were reported on a graph based on two parameters (σ = f(τ)) introduced by Patte and Laffort [34]. σ reflected the ratio between the perceived intensity of the mixture and the sum of the perceived intensities of its components prior to mixing and reflected the level of interaction: σ = Imix/(I_A_ + I_B_). I_AB_ was the overall perceived odor intensity of the mixture of A and B; IA and IB were the perceived odor intensity of A and B components smelled alone. Tau (τ) represented the relative proportion of perceived intensity of A or B unmixed odorant in the binary mixture: τ_A_ = I_A_/(I_A_ + I_B_) or τ_B_ = I_B_/(I_A_ + I_B_).

The graph was divided into five parts according to the interaction level. The position of experimental points reflects the interaction level. The intensity may be as strong as the sum of the perceived intensities of the unmixed components, exemplifying complete addition (σ = 1). The intensity may be also more intense than the sum of its components, exemplifying hyper-addition (σ > 1), or less intense than the sum of its components, exemplifying hypo-addition (σ < 1). In addition, hypo-addition was divided in three different subtypes: “partial addition”, “compromise”, and “subtraction”. They are used if the quality intensity of the mixture is greater than, intermediate to, or smaller than that of the individual compounds, respectively. For each sample, the significance of the observed perceptual interaction was statistically tested by calculating the 95% confidence interval on the mean intensity of the 15 subjects for both σ and τ [9].

#### 3.7.6. Factorial Design

The panel was recruited for the evaluation of the mixtures. The 24 factorial design was constructed with the four most important odor notes (fruity, sour, green, and woody) identified from the omission tests. This design thus involved 16 mixtures corresponding to all possible combinations of zero to four odor notes added to the RJMS. Participants rated the odor typicality answering the question: “According to you, is that a good or bad example of a caramel odor?” on a 10 cm unstructured linear scale labeled with “very bad example” at the left anchor and “very good example” at the right anchor. The other detailed factorial design analysis referred to the previous study [14].

### 3.8. Data Analysis

The aroma intensity, concentration of volatile compounds, and sensory analysis were submitted to analysis of variance (ANOVA). Duncan’s multiple comparison tests were applied to determine significant differences using XLSTAT ver.7.5 (Addinsoft, New York, NY, USA).

## 4. Conclusions

GC–MS, GC–O analysis and sensory evaluation were successfully used to investigate aroma compounds of RRT samples. A total of 37 compounds were identified according to GC–O analysis. These volatile compounds were quantitated according to the GC–MS data. Furthermore, 23 compounds were detected as important odorants according to their OAVs. Additionally, the omission tests confirmed that fruity, sour, green, and woody were the key notes for the aroma of RRT sample. The phenomenon of interactions among different notes was also evaluated in this experiment. The result noted that a synergistic effect was occurred between fruity and woody notes.

The specific feature of the RRT odor is based on the composition of the volatile fraction, which contains both odorants exhibiting RRT notes and a large range of odorants with various qualitative properties. Assessing the typicality of the odor is a direct and efficient way to study the impact of odor categories. It is a straightforward evaluation of the positive or negative impact of the category. The woody note has a positive effect when the fruity note is also in the mixture but tends to show a negative effect otherwise. This study showed that all the notes contribute to the RRT typicality due to a complex balance between fruity, sour, green, and woody and RRT notes arising from the presence of esters, carboxylic acids, aldehydes, ketones, and carbocyclic compounds.

## Figures and Tables

**Figure 1 molecules-26-06202-f001:**
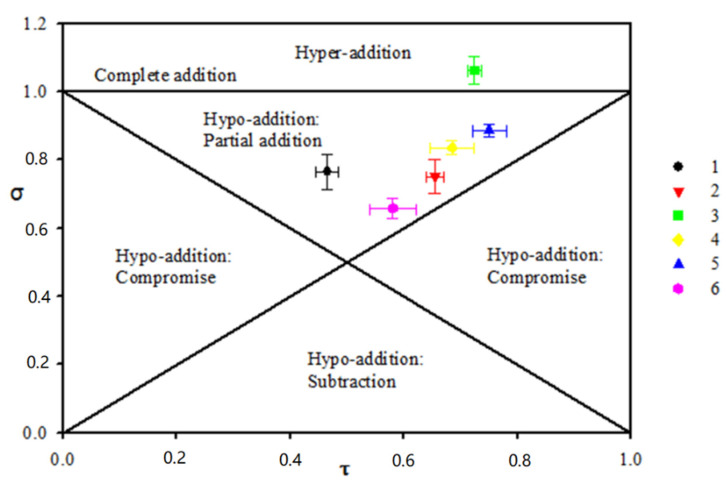
The σ/τ plot of 6 different binary mixtures. 1, Fruity and Sour. 2, Fruity and green. 3, Fruity and woody. 4, Sour and green. 5, Sour and woody. 6, Woody and green.

**Figure 2 molecules-26-06202-f002:**
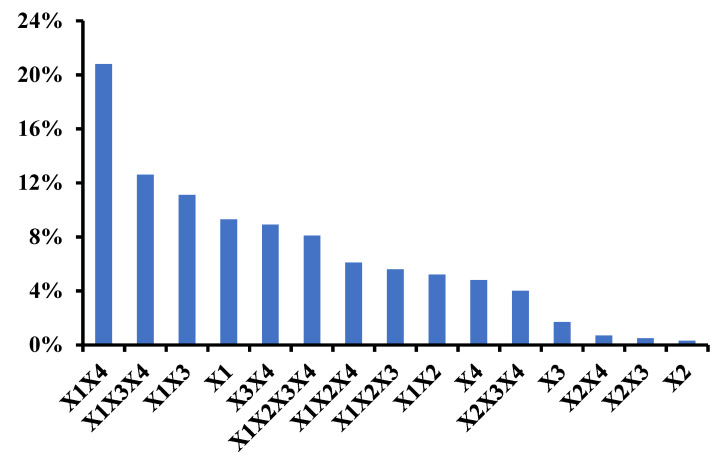
Pareto chart of the factors and interactions of the 24 factorial design (X1 = fruity; X2 = sour; X3 = green; X4 = woody).

**Figure 3 molecules-26-06202-f003:**
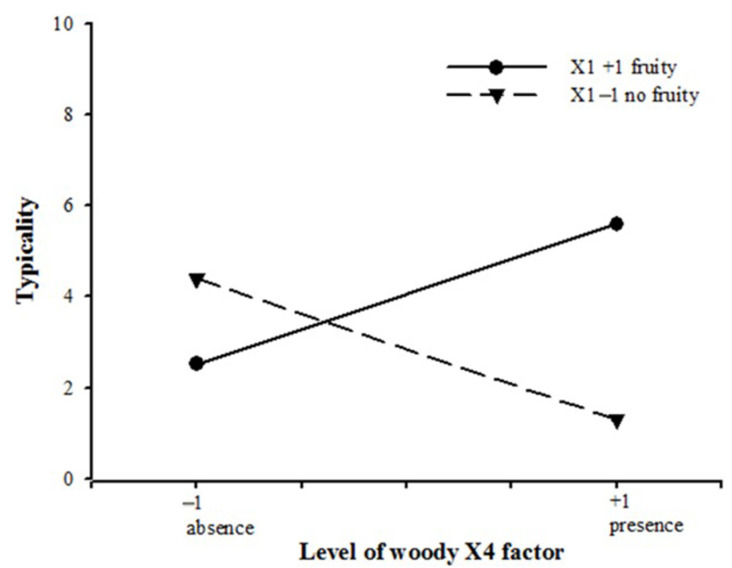
Representation of the interaction between fruity and woody notes (X1X4).

**Table 1 molecules-26-06202-t001:** (**a**) GC–O identified odor-active compounds in RRT samples with the method of aroma intensity. (**b**) Identification, standard curves and concentration (mg/kg) of compounds detected in RRT samples.

(a)									
Code	Compound		RI ^a^	RI ^b^	Identification ^c^		Aroma Intensity ^d^	Odor Description	
1	Ethyl acetate		918	653	AD, RI, Std		3.1	fruity	
2	Ethyl 2-methylpropanoate		992	788	AD, RI, Std		4.3	fruity	
3	Ethyl butanoate		1062	829	AD, RI, Std		4.9	fruity	
4	Ethyl 2-methylbutyrate		1073	877	AD, RI, Std		4.2	fruity	
5	3-Methylbutyl acetate		1142	901	AD, RI, Std		3.3	fruity, banana	
6	2-Heptanone		1201	916	AD, RI, Std		1.5	fatty	
7	3-Methyl-1-butanol		1233	765	AD, RI, Std		2.4	fatty	
8	E-2-hexenal		1241	880	AD, RI, Std		3.9	green, leaf	
9	Ethyl hexanoate		1254	1026	AD, RI, Std		4.6	fruity	
10	Ethyl tiglate		1259	967	AD, RI, Std		1.6	fruity	
11	3,7-Dimethyl-1,3,6-Octatriene		1269	1077	AD, RI, Std		3.0	green	
12	Phenyl ethylene		1281	917	AD, RI, Std		1.9	floral	
13	Ethyl 3-hexenoate		1327	1034	AD, RI, Std		2.2	fruity	
14	2-Heptanol		1342	927	AD, RI, Std		3.0	fatty	
15	Ethyl heptanoate		1356	1125	AD, RI, Std		2.4	fruity	
16	Ethyl E-2-hexenoate		1370	1072	AD, RI, Std		3.3	fruity, green	
17	Hexanol		1375	897	AD, RI, Std		2.5	fatty	
18	2-Nonanone		1412	1119	AD, RI, Std		2.4	fatty	
19	Nonanal		1418	1133	AD, RI, Std		2.2	fatty	
20	Etheyl octanoat		1459	1226	AD, RI, Std		3.8	fruity	
21	Acetic acid		1479	641	AD, RI, Std		2.9	sour	
22	2-Nonanol		1542	1132	AD, RI, Std		2.6	fatty	
23	Benzaldehyde		1561	987	AD, RI, Std		2.4	nutty	
24	2,6,6,10-Tetramethyl-1-oxaspiro(4.5)dec-9-ene		1576	1344	AD, RI, Std		2.6	tea, woody	
25	2-Undecanone		1629	1326	AD, RI, Std		1.9	fruity, green	
26	Caryophyllene		1638	1472	AD, RI, Std		2.2	woody	
27	Butyric acid		1659	818	AD, RI, Std		2.6	sour	
28	3-Methylbutanoic acid		1700	875	AD, RI, Std		3.2	sour	
29	Ethyl benzoate		1707	1202	AD, RI, Std		2.7	floral	
30	Hexanoic acid		1880	1035	AD, RI, Std		4.6	sour	
31	α-Ionone		1899	1532	AD, RI, Std		1.8	floral, woody	
32	α-Iononol		1940	1425	AD, RI, Std		1.7	floral, woody	
33	E-3-hexenoic acid		1980	1053	AD, RI, Std		2.7	sour, fruity	
34	Heptanoic acid		1989	1109	AD, RI, Std		3.0	sour	
35	Octanoic acid		2110	1213	AD, RI, Std		1.8	sour	
36	Ethyl cinnamate		2195	1508	AD, RI, Std		2.0	floral	
37	Eugenol		2227	1397	AD, RI, Std		1.7	woody, floral	
(**b**)									
**Code**	**Compound**	**Identification ^e^**	**Standard Curves ^f^**		**Range ^g^**	**R^2^**	**Concentration** **(mg/kg)**	**TH Literature (mg/kg) ^h^**	**OAV**
1	Ethyl acetate	MS, RI, Std	y = 0.0047x + 0.0002		0.17–63.85	0.997	45.24 ^i^ ± 2.02 ^j^	3.3	14
2	Ethyl 2-methylpropanoate	MS, RI, Std	y = 0.0791x − 0.0001		0.00069–0.27	0.994	0.12 ± 0.01	0.0001	1167
3	Ethyl butanoate	MS, RI, Std	y = 0.1131x + 0.0004		0.0050–2.01	0.999	0.59 ± 0.00	0.00018	3279
4	Ethyl 2-methylbutyrate	MS, RI, Std	y = 0.2248x − 0.0011		0.0040–1.61	0.997	0.24 ± 0.00	0.0003	811
5	3-Methyl butylacetate	MS, RI, Std	y = 0.2804x + 0.005		0.0044–1.77	0.999	0.19 ± 0.01	0.005	38
6	2-Heptanone	MS, RI, Std	y = 0.2183x + 0.0056		0.0025–0.98	0.993	0.12 ± 0.00	0.14	<1
7	3-Methyl-1-butanol	MS, RI, Std	y = 0.0135x − 0.0001		0.0081–3.24	1.000	2.67 ± 0.14	1	3
8	E-2-hexenal	MS, RI, Std	y = 0.0517x + 0.0046		0.0073–2.90	0.991	1.78 ± 0.09	0.082	22
9	Ethyl hexanoate	MS, RI, Std	y = 0.3425x + 0.185		0.071–28.21	0.993	2.21 ± 0.10	0.001	2205
10	Ethyl tiglate	MS, RI, Std	y = 0.3065x + 0.0087		0.002–0.81	0.991	0.06 ± 0.00	0.065	<1
11	3,7-Dimethyl-1,3,6-octatriene	MS, RI, Std	y = 0.0535x − 0.001		0.0019–0.76	0.992	0.49 ± 0.01	0.034	14
12	Phenyl ethylene	MS, RI, Std	y = 0.3585x − 0.0023		0.0021–0.84	0.995	0.08 ± 0.00	0.065	1
13	Ethyl 3-hexenoate	MS, RI, Std	y = 0.5282x + 0.0055		0.0014–0.57	0.990	0.03 ± 0.00	0.25	<1
14	2-Heptanol	MS, RI, Std	y = 0.1362x + 0.0151		0.012–4.80	0.996	1.06 ± 0.08	0.081	13
15	Ethyl heptanoate	MS, RI, Std	y = 0.6947x − 0.0037		0.00098–0.40	0.995	0.02 ± 0.00	0.002	12
16	Ethyl E-2-hexenoate	MS, RI, Std	y = 0.6697x + 0.0035		0.0021–0.83	1.000	0.04 ± 0.00	0.00119	30
17	Hexanol	MS, RI, Std	y = 0.0157x + 0.001		0.027–10.6	0.995	2.20 ± 0.10	1.6	1
18	2-Nonanone	MS, RI, Std	y = 0.7779x − 0.0003		0.0030–1.19	1.000	0.05 ± 0.00	0.082	<1
19	Nonanal	MS, RI, Std	y = 0.2457x + 0.0063		0.00092–0.37	0.997	0.02 ± 0.00	0.04	<1
20	Etheyl octanoat	MS, RI, Std	y = 0.5162x − 0.0779		0.015–6.09	0.993	0.54 ± 0.02	0.015	36
21	Acetic acid	MS, RI, Std	y = 0.0014x + 0.001		0.28–112.81	0.992	88.72 ± 3.03	26	3
22	2-Nonanol	MS, RI, Std	y = 0.5774x + 0.0702		0.014–5.52	0.999	0.20 ± 0.01	0.082	2
23	Benzaldehyde	MS, RI, Std	y = 0.1418x + 0.0056		0.0036–1.45	0.994	0.30 ± 0.01	3.5	<1
24	2,6,6,10-Tetramethyl-1-oxaspiro(4.5)dec-9-ene	MS, RI, Std	y = 0.4911x − 0.0095		0.0032–1.28	0.997	0.11 ± 0.01	0.1	1
25	2-Undecanone	MS, RI, Std	y = 0.8446x − 0.0066		0.0011–0.44	0.996	0.03 ± 0.00	0.082	<1
26	Caryophyllene	MS, RI, Std	y = 0.0853x − 0.013		0.0031–1.25	0.991	0.64 ± 0.03	1.5	<1
27	Butyric acid	MS, RI, Std	y = 0.1664x + 0.0143		0.0019–0.76	0.997	0.07 ± 0.00	1.4	<1
28	3-Methylbutanoic acid	MS, RI, Std	y = 0.0081x + 0.0005		0.053–21.26	0.999	8.69 ± 0.14	0.25	35
29	Ethyl benzoate	MS, RI, Std	y = 0.7402x + 0.0903		0.012–4.96	0.996	0.10 ± 0.01	0.06	2
30	Hexanoic acid	MS, RI, Std	y = 0.0267x + 0.0036		0.38–155.80	0.994	97.12 ± 4.32	1.8	54
31	α-Ionone	MS, RI, Std	y = 1.4384x − 0.0186		0.0015–0.59	0.993	0.03 ± 0.00	0.0027	10
32	α-Iononol	MS, RI, Std	y = 0.9664x − 0.0453		0.0067–2.69	0.995	0.14 ± 0.01	Unknown	-
33	E-3-hexenoic acid	MS, RI, Std	y = 0.1469x − 0.0025		0.0013–0.53	0.992	0.14 ± 0.01	Unknown	-
34	Heptanoic acid	MS, RI, Std	y = 0.1257x − 0.0374		0.0057–2.29	0.993	0.89 ± 0.02	0.91	<1
35	Octanoic acid	MS, RI, Std	y = 0.2847x − 0.0094		0.00063–0.25	0.992	0.06 ± 0.00	1.9	<1
36	Ethyl cinnamate	MS, RI, Std	y = 0.7075x − 0.0592		0.011–4.56	0.994	0.30 ± 0.02	0.04	7
37	Eugenol	MS, RI, Std	y = 0.2918x − 0.004		0.0013–0.52	0.996	0.07 ± 0.00	0.15	<1

^a^ Retention indices of unknown compounds on INNOWAX capillary column. ^b^ Retention indices of unknown compounds on DB-5 Column. ^c^ RI: retention index, Std: confirmed by authentic standards, AD: aroma descriptor. ^d^ The mean aroma intensity which was evaluated by panelists of triplicates according to GC–O analysis. ^e^ Method of identification: MS, mass spectrum comparison using Wiley library; RI, retention index in agreement with literature value; Std, confirmed by authentic standards. ^f^ y is the ratio of the area of the peak of an authentic standard to that of the internal standard; x is ratio of the concentration of the authentic standard chemical to that of the internal standard. ^g^ The concentration range for plotting a standard curve (mg/kg). ^h^ The threshold of volatile compounds referred to in the literature. ^i^ The average concentrations of triplicates (mg/kg). ^j^ Mean standard deviation (average of triplicate).

**Table 2 molecules-26-06202-t002:** Omission tests from complete TAR.

	Fruity	Sour	Green	Floral	Woody	Fatty	N ^a^	Difference Observed
Complete TAR in RJMS	x	x	x	x	x	x		
Test1	-	x	x	x	x	x	14	***
Test2	x	-	x	x	x	x	14	***
Test3	x	x	-	x	x	x	10	**
Test4	x	x	x	-	x	x	5	=
Test5	x	x	x	x	-	x	11	**
Test6	x	x	x	x	x	-	7	=

***, 0.1% significant level; **, 1% significant level; =, no significant difference; ^a^ N, The number of panel members who correctly passed the aroma difference test.

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
