# Peer review of "Characterization of Volatile Compounds of Rosa roxburghii Tratt by Gas Chromatography-Olfactometry, Quantitative Measurements, Odor Activity Value, and Aroma Intensity"

_molecules, 2021, doi:10.3390/molecules26206202_

Round 1

Reviewer 1 Report

Title: "Characterization of Volatile Compounds of Rosa Roxburghii Tratt by Gas Chromatography-Olfactometry, Quantitative Measurements, Odor Activity Value and Aroma Intensity".

The main purpose of this study was to comprehensively analyze the volatile compounds of Rosa Roxburghii Tratt (RRT) and explore the perceptual interaction between key notes.

I think it is an interesting work and worth for publication. However, it still requires a few improvements which are listed as below.

Line77, please delete the "." after the titles of all levels, the same below.

Line 96, although SPME is a very mature technology in food analysis, it does not perform well in the analysis of acidic compounds. Therefore, the detection of acidic compounds in this study was not sufficient.

Line 98-100, the judgment conditions for the odor-active compounds that have the greatest impact on the aroma of RRT fruits should be given.

Line119, the expression of the concentration of each compound here is different from the mg/kg in Table 1b. This question has appeared many times throughout the text. Please unify all of this.

Line 138-139, please add references to support this conclusion.

Line 143-145, OAVs are a measure for evaluating the contribution of a compound in a sample. The expression of this sentence seems to express this meaning, but there is a language problem, please check and correct it.

Line 201, there is no text explanation about Figure 2 found throughout the article, please add it or mark it clearly in the article.

Line 226-228, the ripeness of the RRT fruit seems to be ignored here, it should be explained clearly, which has an important influence on the aroma research of the fruit.

Line 249-250, please add the length between the Y splitter and the sniffing port.

Line 251-252, please give detailed specifications of the column, such as country, state, etc. Please refer to line 272.

Line 264, the sniffing time of each panelist does not exceed 30 minutes, and the entire analysis and detection process is longer than this time. Will it be another member after 30 minutes? There may be differences between different members, how do you control it?

Line 334-337, Please use the subscript correctly, such as IA。

Author Response

Page 1, Abstract: Insert summary information about Rosa Roxburghii Tratt (economic importance, production, etc).

Response: Rosa roxburghii Tratt is an important commercial horticultural crop in China. The related information has been added in the article. Thanks for your advice.

Page 1, Line 20: Keywords: Change keywords by different words from title to expand search system.

Answer: Thanks for your suggestion. The keywords shouldn’t be similar with the title, and they have been changed in another way.

Page 1, Line 23: The sentence “key indicator for assessing food quality” The sentence is generic. Other factors are just as important as aroma in determining food quality. Authors must modify the sentence.

Answer: There are many factors that are important to assessing food quality. So this sentence has been modified.Thanks for your advice.

Page 1, Line 34-38: Highlight other data referring to Rosa Roxburghii Tratt, including annual production, consumption, among other factors that justify the study.

Answer: The related data has been added in the article, thanks for your suggestion.

Page 1, Line 28-30: The sentence is identical to that verified in the work of Wang et al., 2018. Revise the sentence to avoid similarity.

Answer: The duplication of this sentence has been reduced, thanks for your comment.

Page 1, Line 39: Indicate in which foods GC-O is widely used.

Answer: Relevant fruit information has been added.

Page 2, Line 45-50 and 51-59: The introduction should contain information relevant to a general understanding of the content of the work. More detailed information should be used to support the results throughout the discussion. Review the introduction and add general information about the topic.

Answer: Thanks for your advice. I will read my introduction carefully and improve it to make it more relevant to the topic.

Page 2, Line 50-51: In the sentence “All odor pairs exhibited asymmetry in our study and almost one half (48%) showed compromise level.”: The authors present their own results in the introduction. The intention was not clear. If it is a result of the article presented, include it only in the results and discussion section.

Answer: Thanks for your advice. This sentence belongs to to results part, and it has been deleted in the introduction part and added in discussion section.

Introduction: Insert relevant information about the use of CG-O as a tool to monitor quality, process, changes, among others in juices.

Answer: The relevant information has been added in the section.

Page 2, Line 73-75: The sentence must be removed (similar to the description contained in the guide for authors).

Answer: The sentences have been deleted, thanks for suggestion.

Page 5, Line 115-116: Consider indicating otherwise the indication of mean and standard deviation. The use of superscript letters is similar to what is used in tests of means.

Answer: The letters used in the table will be superscript, which is different from the indication of means in the tests.

Page 6, Line 156-158: Check the term “Na” in the table, as it is different from the legend at the bottom of the table.

Answer: I’m sorry to forget to superscript the “a” and explain it. It has been checked and revised. Thanks for your suggestion

Page 7, Line 189-192: Authors should consider using the information in this sentence in the introduction, due to its theoretical importance.

Answer: This part has been supplemented in the introduction. Thanks for your advice.

Page 9, Line 225 “3.2. Materials”: Highlight the ripening stage of the fruits, as it is a factor that can influence the results.

Answer: The information of the ripening stage of the fruits has been added in the materials parts, thanks for your suggestion.

Page 9, Line 257-259: The information is presented in the sensory analysis item. Remove the sentence.

Answer: The detail of information of the panel has been listed in the sensory analysis and the sentence has been deleted, thanks for your advice.

Other comments:

Material and methods section is well described, requiring few adjustments.

Restructuring the introduction should be considered to provide clearer information

Answer: Related problems in material and methods section have been revised and adjusted, while the instruction part has been revised and reconstructed.

Reviewer 2 Report

Comments:

Page 1, Abstract: Insert summary information about Rosa Roxburghii Tratt (economic importance, production, etc).

Page 1, Line 20: Keywords: Change keywords by different words from title to expand search system.

Page 1, Line 23: The sentence “key indicator for assessing food quality” The sentence is generic. Other factors are just as important as aroma in determining food quality. Authors must modify the sentence.

Page 1, Line 34-38: Highlight other data referring to Rosa Roxburghii Tratt, including annual production, consumption, among other factors that justify the study.

Page 1, Line 28-30: The sentence is identical to that verified in the work of Wang et al., 2018. Revise the sentence to avoid similarity.

Page 1, Line 39: Indicate in which foods GC-O is widely used.

 Page 2, Line 45-50 and 51-59: The introduction should contain information relevant to a general understanding of the content of the work. More detailed information should be used to support the results throughout the discussion. Review the introduction and add general information about the topic.

Page 2, Line 50-51: In the sentence “All odor pairs exhibited asymmetry in our study and almost one half (48%) showed compromise level.”: The authors present their own results in the introduction. The intention was not clear. If it is a result of the article presented, include it only in the results and discussion section.

Introduction: Insert relevant information about the use of CG-O as a tool to monitor quality, process, changes, among others in juices.

Page 2, Line 73-75: The sentence must be removed (similar to the description contained in the guide for authors).

Page 5, Line 115-116: Consider indicating otherwise the indication of mean and standard deviation. The use of superscript letters is similar to what is used in tests of means.

Page 6, Line 156-158: Check the term “Na” in the table, as it is different from the legend at the bottom of the table.

Page 7, Line 189-192: Authors should consider using the information in this sentence in the introduction, due to its theoretical importance.

Page 9, Line 225 “3.2. Materials”: Highlight the ripening stage of the fruits, as it is a factor that can influence the results.

Page 9, Line 257-259: The information is presented in the sensory analysis item. Remove the sentence.

Other comments:

Material and methods section is well described, requiring few adjustments.

Restructuring the introduction should be considered to provide clearer information.

Author Response

Line77, please delete the "." after the titles of all levels, the same below.

Answer: I’m sorry to misunderstand the provided template. The problem has been settled, thanks for advise.

Line 96, although SPME is a very mature technology in food analysis, it does not perform well in the analysis of acidic compounds. Therefore, the detection of acidic compounds in this study was not sufficient.

Answer: Only by using one extraction method can keep the other variables of the experiment constant. In order to highlight the study of acidic compounds, Rosa Roxburghii Tratt can be further studied by other methods in the future.

Line 98-100, the judgment conditions for the odor-active compounds that have the greatest impact on the aroma of RRT fruits should be given.

Answer: 11.The information about the size of AI as the judgment condition has been added, thanks for your advise.

Line119, the expression of the concentration of each compound here is different from the mg/kg in Table 1b. This question has appeared many times throughout the text. Please unify all of this.

Answer: The expression of the concentration have been unified

Line 138-139, please add references to support this conclusion.

Answer: The related reference has been added.

Line 143-145, OAVs are a measure for evaluating the contribution of a compound in a sample. The expression of this sentence seems to express this meaning, but there is a language problem, please check and correct it.

Answer: The language has been corrected, thanks for suggestion.

Line 201, there is no text explanation about Figure 2 found throughout the article, please add it or mark it clearly in the article.

Answer: Each figure and table should explained in the article. The text explanation of figure 2 has been added, thanks for the correction.

Line 226-228, the ripeness of the RRT fruit seems to be ignored here, it should be explained clearly, which has an important influence on the aroma research of the fruit.

Answer: The ripeness of fruits is an important factor to the results of experiment. The ripeness of the fruit has been added in the article, thanks for your correction.

Line 249-250, please add the length between the Y splitter and the sniffing port.

Answer: Sorry, it is my negligence. I have modified it in the specific position of the article.

Line 251-252, please give detailed specifications of the column, such as country, state, etc. Please refer to line 272.

Answer: Thank you for your comments. I am sorry that I omitted this condition and it has been corrected in the article.

Line 264, the sniffing time of each panelist does not exceed 30 minutes, and the entire analysis and detection process is longer than this time. Will it be another member after 30 minutes? There may be differences between different members, how do you control it?

Answer: Thank you for your question. I have added the specific information of the instrument in the specific position of the article.

Line 334-337, Please use the subscript correctly, such as IA。

Answer: Thank you for your comments. I am sorry that I forget to change the format of letters and it has been corrected in the article.
